# Thyroid Autoimmune Disease—Impact on Sexual Function in Young Women

**DOI:** 10.3390/jcm10020369

**Published:** 2021-01-19

**Authors:** Ana-Maria Cristina Bortun, Viviana Ivan, Dan-Bogdan Navolan, Liana Dehelean, Andreea Borlea, Dana Stoian

**Affiliations:** 1PhD School Department, Victor Babes University of Medicine and Pharmacy, 300041 Timisoara, Romania; anamariacristina09@yahoo.com (A.-M.C.B.); borlea.andreea@umft.ro (A.B.); 2Department of Internal Medicine II, Victor Babes University of Medicine and Pharmacy, 300041 Timisoara, Romania; stoian.dana@umft.ro; 3Department of Obstetrics and Gynecology, Victor Babes University of Medicine and Pharmacy, 300041 Timisoara, Romania; navolan.dan@umft.ro; 4Neurosciences Department, Victor Babes University of Medicine and Pharmacy, 300041 Timisoara, Romania; lianadeh@umft.ro

**Keywords:** autoimmune thyroid disease, female sexual dysfunction, FSFI, BDI-II

## Abstract

The important prevalence of autoimmune thyroid disease (AITD) in the general population was the main motivation for conducting the present study. The present paper aims to estimate the possible comorbidities related to female sexual dysfunction (FSD) and depression related to AITD. The study group consisted of 320 patients: 250 cases known with untreated AITD, divided into subgroups (euthyroid subgroup, subclinical hypothyroidism subgroup and clinical hypothyroidism subgroup); respectively 70 healthy females in the control group. Patients underwent thyroid evaluation, ovarian evaluation and laboratory assays. At the time of the diagnosis of autoimmune thyroid disease, psychometric scales were filled in by the patients: the Female Sexual Function Index 6 (FSFI-6) and the Beck’s Depression Inventory-II (BDI-II). It was observed that healthy patients had significantly higher FSFI scores than patients with AITD (28 vs. 27; *p* = 0.006). In the AITD group, the risk of FSD increases with the severity of thyroid disease. The most affected areas were: sexual desire (*p* < 0.001), lubrication (*p* = 0.001) and orgasm (*p* = 0.008), followed by excitability and sexual satisfaction. The severity of hypothyroidism influences the degree of decrease in libido, central and peripheral excitability. Sexual satisfaction and orgasm were less influenced. The field related to pain seems uninfluenced by the presence of thyroid disease. The concomitant presence of depression and the value of thyroid-stimulating hormone (TSH) are risk factors in the development of FSD. Higher TSH value and BDI-II score increase the risk of female sexual dysfunction by 1.083 and 1.295 times, respectively. Our findings are significant and promising; they may help professionals dealing with sexual and reproductive health. Despite the importance of female sexual dysfunction and its prevalence, clinicians and patients often ignore it. In fact, only a small percentage of patients consult their doctors about sexual health, and their doctors do not often ask them questions related to this aspect.

## 1. Introduction

Autoimmune thyroid disease (AITD) is a common disorder that preferentially affects women, of all ages. It may occur as such or be integrated as part of the familial predisposition to autoimmune thyroid disease or autoimmune polyendocrinopathies. The clinical picture is variable with temporary or permanent euthyroidism or hypothyroidism and rarely with transient thyrotoxicosis. Common characteristics are the presence of anti-thyroid peroxidase antibodies and lymphoplasmocytic infiltration of the thyroid parenchyma [1].

The motivation for conducting the present study is that autoimmune thyroid pathology is very common, with a significant interest in a great percentage of the female population, studies in the field claiming an approximate incidence of 350/100,000 women/year in the case of hypothyroidism and about 80/100,000 women/year in the case of hyperthyroidism [2]. The prevalence of this condition increases with age and is favoured by exposure to increased amounts of iodine [3]. Vanderpump and collab. found that the mean incidence (with 95% Confidence Intervals) of spontaneous hypothyroidism in women was 3.5/1000 survivors/year (2.8–4.5) rising to 4.1/1000 survivors/year (3.3–5.0) for all causes of hypothyroidism and in men was 0.6/1000 survivors/year (0.3–1.2). The mean incidence of hyperthyroidism in women was 0.8/1000 survivors/year (0.5–1.4) and was negligible in men [4].

Sexual dysfunction refers to a difficulty in performing normal sexual activities and experiencing physical pleasure, desire, arousal or orgasm. According to the Diagnostic and Statistical Manual of Mental Disorders (DSM-5) published by the American Psychiatric Association (APA), sexual dysfunction refers to extreme suffering during normal sexual activity for at least 6 months, in the absence of any drug-induced substance or sexual dysfunction [5]. FSD includes the disorders of desire, arousal, lubrication, orgasm and sexual satisfaction, as well as the sensation of pain [6]. Thyroid disease is considered a risk factor for sexual dysfunction [7]. Both hypothyroidism [8] and hyperthyroidism were shown to be associated with impairment of desire, arousal, lubrication, orgasm, satisfaction and pain during sexual intercourse in women [9]. The development of sexual dysfunction is attributed to the effect of thyroid hormones on the level of circulating sex hormones through the peripheral and central pathways, indirectly leading to psychiatric and autonomous disorders and causing impairment of sexual function. The correction of the thyroid condition is associated with the resolution of sexual dysfunction in both men and women [10].

Hormones directly or indirectly regulate all sexual functions (desire, arousal, lubrication, orgasm). Some sexual symptoms may occur as a psychosomatic consequence of hormonal impairment. However, in other cases, endocrine insufficiency may be caused by psychosomatic factors. Autoimmune thyroid disease alters central excitability, the phenomenon being independent of the value of thyroid-stimulating hormone (TSH) and being present in both hypothyroidism and hyperthyroidism [11]. AITD associates changes in brain activity, so the TSH value indirectly correlates with central brain flow and conditions overall brain activity. The impact is independent of thyroid disease or depression. The optimal value of TSH for normal brain activity has been shown to be <3 mIU/L [12].

Autoimmune thyroid disease associated with female sexual dysfunction is presented by central symptoms (dramatic decrease in libido, depression, weight gain, lethargy) and peripheral symptoms (alteration of vaginal lubrication as a phenomenon independent of variations in estrogenemia, being a possible psychosomatic effect). It also associates a decrease in sexual satisfaction, anxiety, irritability, difficulty in achieving orgasm and potentially promotes dyspareunia. Vaginismus was not found to be influenced by autoimmune thyroid disease [13].

Hypothyroidism has been observed to induce ovarian failure, so long-term hypothyroidism leads to increased ovarian volume and the formation of cysts [14]. A TSH value below 2.5 mIU/L was associated with normal ovarian function in a 2013 study [15].

Recently, a connection between various autoimmune conditions and psychiatric disorders has been discovered [16]. With the introduction of new immunological techniques and the expansion of immunoneuropsychiatric research, there is growing evidence that at least one subset of psychiatric disorders has an autoimmune ground. In view of these perspectives, our analysis will contribute to further clarification of the association of psychiatric pathology and AITD [17].

Regarding depression, most studies conclude that thyroid autoimmunity is associated with depressive disorder. It is therefore important to raise awareness among physicians about this connection in order to speed up the diagnostic process. In patients with depressive disorders, evaluation for autoimmune thyroiditis should be carried out, and in patients with thyroiditis, screening for psychiatric symptoms is required [18].

The presence of depressive disorder may be linked to the presence of high anti-thyroid peroxidase antibody titres in patients, regardless of thyroid function. However, current data do not show any link between slightly increased levels of anti-thyroid peroxidase antibodies and depressive symptoms in the general population. The relative risk of depression is highest among clinical forms of hypothyroidism and hyperthyroidism (7.9 times, respectively 6.1 times higher), followed by subclinical forms (2.9 times higher in hypothyroidism and 3.3 times higher in the case of hyperthyroidism) and decreases in asymptomatic forms of autoimmune thyroid disease (3 times higher risk) [19].

Sexual dysfunction is highly prevalent among depressive patients and is also a common side effect of treatment with antidepressants. It was also more prevalent in those with a comorbid disease that could affect sexual function and in those who consider sexual activity unimportant [20].

Research on female sexual dysfunction has advanced rapidly in recent years. This revealed the need for robust psychometric tools for diagnosing female sexual dysfunction and for effectively monitoring treatment-induced changes [21].

Female Sexual Function Index (FSFI) is a measuring scale with 19 self-reporting themes of female sexual function, which provides scores for the general aspects of sexual function in women.

Certainly, the availability of an extraordinarily large database of published studies, including results from several observational and interventional studies in different study populations makes the FSFI a highly professional and scientific inventory. The tool is also relatively short, can be completed in 5 min or less and has a simple and practical score system. In addition, the underlying structure of the domain, which was designed to assess sexual desire and arousal separately, in addition to other components of sexual function in women, has been validated and reproduced in several studies [22].

The 6-item FSFI is a short form of the original 19-item FSFI that measures sexual function in women. It is a brief questionnaire comprising the same six domains, allowing quick assessments both in clinical and research settings [23].

Depression is a hugely prevalent pathology globally, with a major impact on the population [24]. Of the self-assessment tools available, Beck’s Depression Inventory-II (BDI-II), a 21-item psychometric scale, is one of the world’s most popular measuring tools for depressive symptoms [25]. First proposed by Beck and his collaborators, this tool has been used in more than 7000 studies so far [26].

Beck’s Depression Inventory—BDI-II is a self-assessment scale for measuring symptoms of depression and their severity [27].

Being a very well-designed questionnaire, investigators benefit from this simple, short, reliable and validated tool to design research in a variety of directions [28].

## 2. Materials and Methods

### 2.1. Establishment of the Study Group

The study group consisted of 320 patients aged 20 to 45 years: 250 cases known with untreated AITD, respectively 70 healthy females in the control group. A total of 250 patients were diagnosed with AITD and were divided into subgroups, according to the level of thyroid dysfunction: euthyroid subgroup (E), comprising 146 euthyroid patients, with TSH and FT4 levels in the normal range; subclinical hypothyroidism subgroup (S), comprising 51 patients with increased TSH values, but FT4 in the normal range; clinical hypothyroidism subgroup (C), comprising 53 patients with increased TSH values and low FT4 values.

### 2.2. Inclusion Criteria and Exclusion Criteria

The study group included sexually active patients in a stable relationship for minimum 12 months, at least 12 months after delivery, with no breastfeeding in the past 6 months, agreeing to the idea of sexual evaluation, with a recent diagnosis of autoimmune thyroid disease and no specific treatment beforehand. Patients with previously known AITD, already under supplemental treatment with thyroid hormones were not included in the study group. Patients with associated ovarian or gynecological pathology, adrenal conditions, diagnosed depression or other psychiatric conditions of axis I, patients who have undergone any type of antipsychotic, antidepressant or anxiolytic treatment, patients with chronic diseases and those with severe myxedema were excluded from the present study. Apparently healthy women, considering the same inclusion and exclusion criteria, constituted the control group.

The study was performed in accordance with the Ethical Guidelines of the Helsinki Declaration and was approved by the University Ethics Committee. All subjects agreed to the evaluation and gave their written informed consent prior to inclusion.

### 2.3. Method

Patients were recruited from the SCJUPBT Outpatient Endocrinology Clinic in Timisoara, Romania over a period of 12 months (1 January 2019 to 31 December 2019).

Patients underwent thyroid evaluation by means of: clinical examination, neck ultrasound and laboratory analysis (TSH, FT4, anti-thyroidperoxidase antibodies and anti-thyroidglobulin antibodies). Ovarian evaluation was performed by anamnesis—quality of menstruation, quantity, periodicity, clinical examination, normal sexualisation, lack of hyperandrogenic signs and laboratory assays performed on the 3rd day of the menstrual cycle: estradiol and prolactin.

The laboratory parameters were analyzed by the immunochemical method with electrochemiluminescent detection (ECLIA). Venous blood samples were collected in the morning, after a fasting interval of a minimum of 8 to 12 h, at least 2 h after the time of awakening. Since prolactin secretion is influenced by circadian rhythm, blood specimens were collected between 8–10 am with minimum rest of 30 min before sampling. Patients were instructed to avoid caffeine-based drinks, smoking, stress and exercise before blood sampling.

For laboratory parameters, the following normal ranges were considered: for TSH 0.4–4.0 mIU/L, for FT4 12.0–22.0 pmol/L, for anti-TPO antibodies < 34 IU/L, for anti-TG antibodies < 115 IU/L, for estradiol 46–607 pmol/L and for prolactin 2.4–24.5 ng/mL.

At the time of the diagnosis of autoimmune thyroid disease, psychometric scales were filled in by the patients: the FSFI-6 and the BDI-II questionnaire.

#### 2.3.1. FSFI-6 (Female Sexual Function Index)

The questionnaire was translated into Romanian by two independent translators and its precision was ascertained through reverse translation by an expert team. A group of volunteers participated in an evaluation of the test–retest reliability of the Romanian version. The data were summarized using descriptive statistics.

The questionnaire was given to patients at the time of AITD diagnosis. It comprises six domains: desire, arousal, lubrication, orgasm, satisfaction and pain. Desire and satisfaction items are rated on a 5-point Likert scale, ranging from 1 to 5, and the other items are rated on a 6-point Likert scale, ranging from 0 to 5.

The individual domain scores and full scale (overall) score of the FSFI can be derived from the computational formula outlined below, in Table 1. For individual domain scores, the sum is made by adding individual scores for the items that comprise the domain and multiply the sum by the domain factor (see below). Adding the six domain scores leads to obtaining the full-scale score. It should be noted that, within the individual domains, a domain score of zero indicates a subject reporting no sexual activity in the past month. The minimum score is 2 and the maximum score is 30. A threshold value of ≤ 26 was established for detecting FSD.

#### 2.3.2. BDI-II (Beck’s Depression Inventory)

We used the Romanian validated version of the questionnaire. The process of translation and adaptation of BDI-II in Romania was carried out in accordance with international norms of adaptation of psychological instruments. The instrument meets the psychometric standards necessary for clinical use, supported by the analysis of its fidelity and validity in Romanian. The questionnaire was given to patients at the time of AITD diagnosis. Each item is allocated a score between 0 and 3 points, the total score is obtained by summing up the points obtained for each item. The maximum score for BDI-II is 63 points. In this study, a threshold value of ≥10 was considered to predict clinical depression.

### 2.4. Statistical Analysis

Continuous variables were presented as mean and standard deviation (SD) or median and interquartile range, and categorical variables were presented as frequency and percentages. The Shapiro–Wilk test was employed to assess the distribution of continuous variables. Descriptive and inferential statistical analysis was performed in order to summarize the characteristics of the study population. To compare groups of patients with or without female sexual dysfunction with each other, the *t*-test and ANOVA test were used, with Tukey post hoc analysis or Mann–Whitney *U* test and Kruskal–Wallis test for variables with non-Gaussian distribution. For the nominal variables, cross-tabulation with Chi-squared test was employed. To find the independent predictive factors for female sexual dysfunction logistic regression was employed. Akaike information criteria (AIC) were used in order to determine the best model. The receiver operating characteristic (ROC) curve was employed to illustrate the identification ability and the thresholds of TSH levels to discriminate between the patients with and without female sexual dysfunction, determined with the Youden’s index.

The Shapiro-Wilk test was used to determine the distribution of data in this study. In the analysis of data with a normal distribution (Gaussian), parametric tests were applied and in the case of data with non-Gaussian distribution, nonparametric tests were applied. Non-Gaussian distribution data are represented using the median, respectively Quartile 25 and 75 (Q25–Q75).

Data were collected and analyzed using SPSS v.26 (Statistical Package for the Social Sciences, Chicago, IL, USA). The results obtained are rendered as tables and figures. A *p*-value of <0.05 was considered to indicate a statistically significant difference, with a 95% confidence interval.

## 3. Results

The general parameters that characterize the studied group were similar to those of the control group (age, BMI, duration of the relationship with the partner). All patients were in the age group 20–45 years, with normal values of BMI and in a stable relationship for a minimum of 12 months.

### 3.1. Hormonal Results

Hormonal parameters were evaluated in the control group versus three groups of AITD patients, with different functional status, as shown in Table 2. Following the application of the Anova test, it was observed that there are no statistically significant differences between the controls and euthyroid patients with AITD (subgroup E) in terms of estradiol and prolactin values, but there are significant differences between the controls and the subclinical (subgroup S) and clinical forms of hypothyroidism (subgroup C), *p* < 0.05.

### 3.2. FSFI Evaluation Results

Following the Shapiro–Wilk test, a *p* < 0.001 was obtained, so the data do not show a normal distribution.

The differences between the AITD group and the control group with respect to the total FSFI score and the FSFI scores on separate domains were highlighted using the Mann–Whitney *U* nonparametric test, as shown in Table 3.

Regarding the total score, statistically significant differences were observed between the median of the control group and the AITD group (28 vs. 27; *p* = 0.006). The following subdomains noted statistically significant differences between the two groups: desire (*p* < 0.001), arousal (*p* = 0.041), lubrication (*p* = 0.001) and orgasm (*p* = 0.008). Mann–Whitney Test was applied. Figure 1 also shows the differences between the AITD groups, regading FSFI total score.

Following the application of the Chi-squared test, no statistically significant differences were observed between the control group and the AITD group in terms of the diagnosis of female sexual dysfunction according to the FSFI scale (*p* = 0.336) as shown in Table 4.

With regard to the comparison between the control group and the patient categories with AITD, as shown in Table 5, significant differences were observed following the application of the Chi-squared test (*p* = 0.032). The prevalence of the FSFI score with positive diagnostic value (≤ 26.55) is higher in group C compared to the other groups.

### 3.3. FSFI Evaluation Results in Different Study Subgroups

To illustrate the differences in terms of the total FSFI score calculated and FSFI scores on separate domains between groups with AITD (E, S, C), the nonparametric Kruskal–Wallis test was used. As Table 6 displays below, significant differences were noted between all three AITD groups and the control group in terms of the separate domains of the FSFI as well as the total FSFI (*p* < 0.001). A higher median was noted for the control group compared to the other studied groups; the lowest median was recorded for group C of patients with AITD.

For the total FSFI score, the ANOVA test revealed significant differences between the averages of the rated groups: E, S and C (*p* < 0.001).

Significant statistic differences were observed between group E and group C in terms of average scores obtained in total FSFI (27.14 vs. 24.98, *p* < 0.001), respectively between group S and Group C (26.37 vs. 24.98, *p* = 0.046). No significant differences were observed between group E and group S (27.14 vs. 26.37, *p* = 0.249).

For the FSFI arousal field, statistically significant differences were obtained only between group E and group C, respectively between group S and group C; no significant differences were observed between group E and group S (*p* > 0.05). Table 7 below displays the detailed FSFI scores for AITD categories. The data show no statistically significant differences between the three groups in terms of the FSFI domain of defined pain (*p* > 0.05).

### 3.4. BDI-II

Comparison between patients with AITD and healthy patients revealed that there are no statistically significant differences between the two groups in terms of diagnosis of depression based on the BDI-II scale following the application of the Chi-squared test (*p* = 0.471).

Please note that the positive diagnosis of depression based on the BDI-II scale is made for a score ≥ 10.

Comparison between categories of patients with AITD and healthy patients in terms of depression diagnosis according to the BDI-II reavealed that there are no statistically significant differences in the distribution of BDI-II score in the investigated groups following the application of the Chi-squared test (*p* = 0.172).

### 3.5. Comparison Between AITD Patients and Healthy Patients Regarding BDI-II Total Scores

When comparing the BDI-II score results for the two main groups (AITD patients and healthy patients) no statistically significant differences were noted.

Concerning the BDI-II scores, the Anova test did not show statistically significant differences between groups (*p* = 0.600), as shown in Table 8.

### 3.6. Independent Risk Factors for the Diagnosis of Female Sexual Dysfunction

In order to highlight the independent risk factors for the diagnosis of female sexual dysfunction, we carried out a logistic regression analysis. The initial model included the following determinants: age, estradiol, prolactin, TSH and BDI-II, with the diagnosis of sexual dysfunction as a dependent variable (total FSFI score: ≤ 26.55, positive diagnosis; > 26.55, negative diagnosis). The results are shown in Table 9.

Higher TSH value and BDI-II score increase the risk of female sexual dysfunction by 1.083 and 1.295 times, respectively. This regression model explains 25.9% (R^2^ = 0.259) of the diagnosis for female sexual dysfunction.

A multivariate logistic regression analysis was used to identify the most significant factors that seem to predict the development of female sexual dysfunction. The regression model was built using the backward method and Akaike information criteria (AIC) were applied to determine the best model. The odds ratio and 95% confidence interval were calculated and adjusted for age. TSH and depression were found to be predictive factors for female sexual dysfunction. The risk of developing this condition increases with higher TSH levels and BDI-II score.

### 3.7. Independent Predictors for Female Sexual Dysfunction

Multivariate linear regression was used, having as a dependent factor the total FSFI score. The linear model included the following as independent factors: age, BDI-II score, estradiol value, prolactin, TSH and ATPO antibodies. The results shown in Table 10 point out that TSH and the BDI-II score were found to independently predict FSD.

R^2^ = 0.597 (59.7% of the total FSFI score variation is explained by this model). The increase in TSH and BDI-II leads to a decrease in the total FSFI score, with the relationship being inversely proportional. The increase in BDI-II or TSH score by 1 unit results in a decrease in FSFI total score by 0.350 and 0.255, respectively.

### 3.8. Threshold Value for TSH Providing Positive Diagnosis of Female Sexual Dysfunction

In order to determine the threshold point of TSH value for predicting the diagnosis of female sexual dysfunction, the ROC curve statistics were used.

According to the Youden index criteria, the discriminating point for the diagnosis of female sexual dysfunction is 2.75 for TSH. Thus, the values below this threshold have predictive value for a negative diagnosis of female sexual dysfunction, and the values above 2.75 have predictive value for a positive diagnosis of sexual dysfunction. The AUROC parameters are detailed below in Table 11 and Figure 2.

## 4. Discussion

The present study included patients diagnosed with AITD, who were classified into three groups: E (euthyroid), S (subclinical) and C (clinical). Comparing the control group with patients with AITD in terms of estradiol values, significantly lower values were observed in the AITD group. Within the AITD group, euthyroid patients (group E) had significantly higher serum estradiol values compared to patients with subclinical AITD (group S) and those with clinically manifest AITD (group C). Prolactin levels were found to be higher the more severe the form of AITD, so that healthy patients had significantly lower values than patients with AITD; out of these, the highest prolactin levels were recorded in the clinically diseased group (group C).

These conclusions support studies already published in the field. Hyperprolactinemia has been found to be associated with numerous diseases with autoimmune substrates, such as rheumatoid arthritis, systemic lupus erythematosus, multiple sclerosis; it is considered to have a crucial role in the pathogenesis of these diseases. However, a direct correlation between prolactin levels and the activity of these diseases has not been established [29]. AITD and polycystic ovary syndrome were found to be two acutely related pathological entities, thus explaining significantly lower estradiol levels in thyroid disease [30].

Our study points out and attempts to explain the discordance between similar estradiol and prolactin values in the two groups, but significant differences in FSFI scores, both for the total score and separate fields.

It was observed that healthy patients had significantly higher scores than patients with AITD. In the AITD group, the risk of sexual dysfunction increases with the severity of the thyroid disease.

Scores obtained by patients with a clinically manifest form of disease were significantly lower than those of patients with subclinical forms. A positive score for FSD (score ≤ 26.55) was significantly higher in the C group of patients, those with clinical form of AITD (49.1%), compared to the other groups (29.5% for group E, 27.5% for group S and 27.1% for the control group). Additionally, group C expressed less positive results (>26.55 points) in FSFI (50.9%), compared to the other groups (70.5% for group E, 72.52% for group S and 72.9% for the control group). These results express in more detail the conclusions of the studies carried out by Krysiak and his collaborators in this regard. The results of these studies suggest that excessive thyroid hormone production and thyroid autoimmunity have an additive effect and may negatively affect mood and sexual function in females [31,32].

Regarding the differences between the AITD group (all functional forms) and the control group in terms of FSFI results by domain, there are statistically significant differences for the areas: desire, arousal, lubrication and orgasm. Patients suffering from AITD had lower scores in these domains, arousal being the least affected. Pain was found not to be influenced by the presence of AITD, unlike previous studies claim [33].

It might seem strange that the pain domain is not affected, while arousal is. The explanation is that even if there are some alterations in arousal, they are not major, so they are not able to influence the pain domain. Lubrication is not entirely absent.

For the FSFI arousal field, statistically significant differences were obtained only between group E and group C, respectively between group S and group C, and no significant differences were observed between group E and group S. Arousal decreases as the thyroid is more affected.

Between the three AITD groups (E, S, C) and the control group, in terms of the separate domains of the FSFI, as well as the total FSFI, the median of the control group was higher compared to the other groups, and the lowest median was recorded within group C of patients with AITD.

In terms of the association of depression, in patients with AITD compared to healthy patients, there were no significant differences in the score obtained from the application of the BDI-II scale.

Cayan and collaborators [34] detected the presence of a lower educational level, unemployment, chronic diseases, multiparity and menopause as important risk factors for FSD. Hypothyroidism is also associated with female metabolic syndrome, which promotes the development of cardiovascular disease and type 2 diabetes, both of which independently lead to the development of FSD [35]. Additionally, Maseroli and collaborators [36] found that clitoral vascular resistance is positively associated with decreased sexual arousal, body image concerns and increased somatized anxiety symptoms. Given the complexity of the results of the above-mentioned studies, it is worth designing a good prospective control study in the future.

One of the aspects to be considered in clinical practice, during the monitoring of this category of patients diagnosed with AITD, refers to the independent risk factors associated with a possible diagnosis of female sexual dysfunction. In our study, it was found that the risk of female sexual dysfunction increases both with the increase in serum TSH, as well as with the increase in BDI-II score. In the case of TSH, the risk increases 1.083 times with each unit of increase in TSH, and in the case of the BDI-II score, the risk increases 1.295 times with each point obtained from the application of the scale.

It was also noted that a threshold value of 2.75 mIU/L predicts FSD with a sensitivity of 62.7%, a specificity of 57.8% and an accuracy of 58.75% [AUROC 0.592 (0.527–0.657)]. Thus, values below this threshold have a relatively good predictive value for a negative diagnosis of FSD. These findings complete the data obtained by other studies in the field, showing that TSH has a notable role in the development of FSD [37].

The present study is not without some limitations, the sample size of the cases included in the study being small.

In view of the above, the investigation of sexual dysfunction should be part of the diagnostic process, which can help doctors from different clinical backgrounds to assess the sexual problems of female patients with specific clinical conditions, so as to facilitate their identification and possible treatment. 

## 5. Conclusions

The presence of thyroid disease in the female population is associated with sexual dysfunction, so that the incidence of sexual dysfunction, defined objectively by the value of the total FSFI score ≤ 26.55 points, is significantly higher in the group of patients with autoimmune thyroiditis, regardless of the stage of the condition.

The most affected areas, in terms of frequency, are: sexual desire, lubrication and orgasm, followed by (the field of) excitability and implicitly that of sexual satisfaction. The severity of hypothyroidism influences the degree of decrease in libido, central and peripheral excitability. Sexual satisfaction and orgasm are less influenced in this respect. The field related to pain seems uninfluenced by the presence of thyroid disease.

The concomitant presence of depression has a significant impact on the risk of developing female sexual dysfunction.

The value of TSH is also an independent risk factor in the development of sexual dysfunction. The threshold value for TSH with discriminatory value for the diagnosis of sexual dysfunction was found to be close to that considered optimal for a quality central brain flow.

The increased prevalence of sexual pathology in patients with autoimmune thyroiditis is independent of the presence of hyperprolactinemia, respectively of estrogen fluctuations.

Our findings are significant and promising; they may help professionals dealing with sexual and reproductive health. Despite the importance of female sexual dysfunction and its prevalence, clinicians and patients often ignore it. In fact, only a small percentage of patients consult their doctors about sexual health, and their doctors do not often ask them questions related to this aspect.

## Figures and Tables

**Figure 1 jcm-10-00369-f001:**
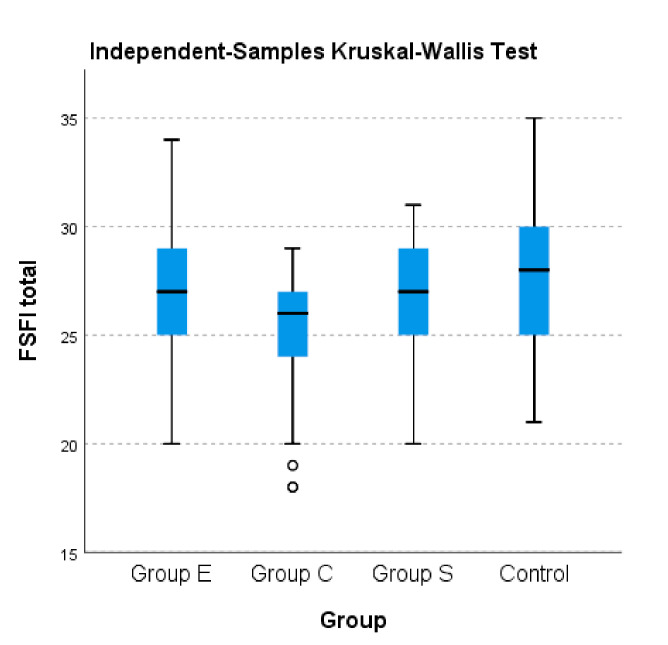
Kruskal–Wallis Test.

**Figure 2 jcm-10-00369-f002:**
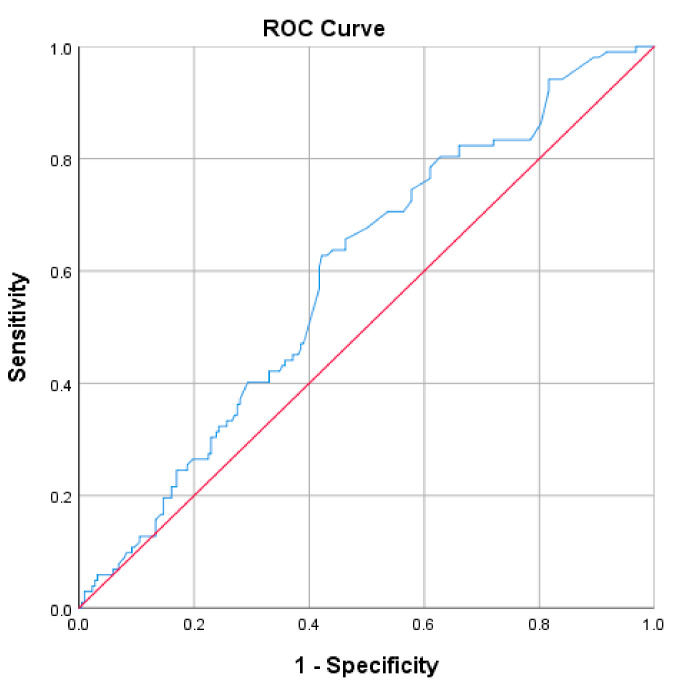
Receiver operating characteristic (ROC) curve for thyroid-stimulating hormone (TSH).

**Table 1 jcm-10-00369-t001:** Female Sexual Function Index-6 (FSFI-6) Scoring Appendix.

Domain	Score Range	Factor	Minimum Score	Maximum Score
Desire	1–5	0.6	2	30
Arousal	0–5	0.3		
Lubrication	0–5	0.3		
Orgasm	0–5	0.4		
Satisfaction	1–5	0.4		
Pain	0–5	0.4		
Full-Scale Score Range				

**Table 2 jcm-10-00369-t002:** Comparison between different autoimmune thyroid disease (AITD) groups and healthy patients in terms of hormonal parameters.

	Control Group *n* = 70	E *n* = 146	S *n* = 51	C *n* = 53	*p*-Value *
Estradiol (pmol/L)	157.06 ± 46.920	141.64 ± 34.090	135.39 ± 36.321	138.11 ± 30.070	0.005
Prolactin (ng/mL)	17.936 ± 3.540	17.496 ± 3.934	18.924 ± 2.937	24.2020 ± 3.069	<0.001

* Anova Test; E, euthyroid AITD cases; S, subclinical hypothyroidism; C, clinical hypothyroidism.

**Table 3 jcm-10-00369-t003:** Comparison between AITD group and healthy patients in terms of FSFI results.

	AITD Group *n* = 250	Control Group *n* = 70	FSFI Maximum Score	*p*-Value
FSFI total	27 (25–29)	28 (25–30)	30	0.006 *
FSFI desire	5 (4–5)	5 (5–5)	5	<0.001*
FSFI arousal	5 (5–6)	5 (5–6)	5	0.041 *
FSFI lubrification	5 (4–5)	5 (4.75–5)	5	0.001 *
FSFI orgasm	5 (4–6)	5 (5–6)	5	0.008 *
FSFI satisfaction	5 (4–5)	5 (4–5)	5	0.274
FSFI pain	3 (2–4)	4 (3–5)	5	0.063

* Mann–Whitney *U* Test. Abbreviations: AITD, autoimmune thyroid disease; FSFI, Female Sexual Function Index.

**Table 4 jcm-10-00369-t004:** The distribution of patients (%) with positive diagnosis of female sexual dysfunction according to the FSFI scale (score ≤ 26.55).

	FSFI Positive	FSFI Negative	*p*-Value *
AITD Group	83 (33.2%)	167 (66.8%)	0.336
Control Group	19 (27.1%)	51 (72.9%)	

* Chi-squared Test.

**Table 5 jcm-10-00369-t005:** Comparison between the categories of patients with AITD and healthy patients in terms of diagnosis of female sexual dysfunction according to the FSFI scale (percentage expression).

		FSFI Positive	FSFI Negative	*p*-Value *
AITD Group	E	43 (29.5%)	103 (70.5%)	0.032
	C	26 (49.1%)	27 (50.9%)	
	S	14 (27.5%)	37 (72.52%)	
Control Group		19 (27.1%)	51 (72.9%)	

* Chi-squared Test; E, euthyroid AITD cases; S, subclinical hypothyroidism; C, clinical hypothyroidism.

**Table 6 jcm-10-00369-t006:** Comparison between categories of AITD and healthy patients in terms of FSFI scores by separate domains.

	AITD Group *n* = 250			Control Group *n* = 70	*p*-Value
	E *n* = 146	S *n* = 51	C *n* = 53		
FSFI Total	27 (25–29)	27 (25–29)	26 (24–27)	28 (25–30)	<0.001 *
FSFI Desire	5 (5–5)	4 (4–5)	4 (3–4)	5 (5–5)	<0.001 *
FSFI Arousal	5 (5–6)	5 (5–6)	5 (4–5)	5 (5–6)	<0.001 *
FSFI Lubrication	5 (4–5)	5 (4–5)	4 (4–4)	5 (4.75–5)	<0.001 *
FSFI Orgasm	5 (5–6)	5 (4–5)	4 (4–5)	5 (5–6)	<0.001 *
FSFI Satisfaction	5 (4–6)	5 (4–5)	4 (4–4)	5 (4–5)	<0.001 *
FSFI Pain	3 (3–4)	4 (2–5)	3 (2–4)	4 (3–5)	0.033 *

* Kruskal–Wallis Test; E, euthyroid AITD cases; S, subclinical hypothyroidism; C, clinical hypothyroidism.

**Table 7 jcm-10-00369-t007:** Comparison of categories of AITD in terms of FSFI scores by separate domains.

	Group E *n* = 146	Group S *n* = 51	Group C *n* = 53	*p*-Value		
				E-S	E-C	S-C
FSFI Desire	5 (5–5)	4 (4–5)	4 (3–4)	<0.001 *	<0.001 *	<0.003 *
FSFI Arousal	5 (5–6)	5 (5–6)	4 (4–5)	0.830	<0.001 *	0.025 *
FSFI Lubrication	5 (4–5)	5 (4–5)	4 (4–4)	<0.009 *	<0.001 *	0.020 *
FSFI Orgasm	5 (5–6)	5 (4–5)	4 (4–5)	0.007 *	<0.001 *	0.016 *
FSFI Satisfaction	5 (4–6)	5 (4–5)	4 (4–4)	<0.002 *	<0.001 *	0.035 *
FSFI Pain	3 (3–4)	4 (2–5)	3 (2–4)	-	-	-

* Mann–Whitney *U* test, *p*-value adjusted for pairwise comparison; E, euthyroid AITD cases; S, subclinical hypothyroidism; C, clinical hypothyroidism

**Table 8 jcm-10-00369-t008:** Comparison between the categories of patients with AITD and patients who are healthy in terms of Beck’s Depression Inventory-II (BDI-II) score.

	Control	E *n* = 146	S *n* = 51	C *n* = 53	*p*-Value *
BDI-II Score	8.46 ± 3.999	8.77 ± 5.182	8.88 ± 4.607	7.85 ± 3.207	0.600

* Anova Test. Abbreviations: BDI, Beck’s Depression Inventory; E, euthyroid AITD cases; S, subclinical hypothyroidism; C, clinical hypothyroidism.

**Table 9 jcm-10-00369-t009:** Logistic regression for the determining risk factors.

	OR	95% CI		*p*-Value
TSH	1.083	1.006	1.166	0.034
BDI-II score	1.295	1.190	1.411	< 0.001

Abbreviations: TSH, thyroid stimulating hormone; OR, odds ratio; CI, confidence interval.

**Table 10 jcm-10-00369-t010:** Linear regression for predicting female sexual dysfunction (FSD).

Coefficient	B	SE	*p*	95% Confidence Interval	
BDI-II Score	−0.350	0.031	0.000	−0.412	−0.288
TSH	−0.255	0.042	0.000	−0.338	−0.173

Abbreviations: B, regression coefficient; SE, standard error.

**Table 11 jcm-10-00369-t011:** ROC curve parameters for TSH in relation to the diagnosis of FSD and the impact on female sexual function.

	TSH
AUROC	0.592 (0.527–0.657)
Sensitivity	62.7%
Specificity	57.8%
Positive predictive value	41%
Negative predictive value	76.8%
Accuracy	58.75%
*p*-value	0.008
TSH Threshold Value	2.75

Abbreviations: AUROC, area under the ROC curve; TSH, thyroid stimulating hormone.

## Data Availability

The data presented in this study are available on request from the corresponding author. The data are not publicly available due to reasons concerning privacy of the subjects.

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
