# Peer review of "Thyroid Autoimmune Disease—Impact on Sexual Function in Young Women"

_jcm, 2021, doi:10.3390/jcm10020369_

Round 1
Reviewer 1 Report
Thank you for the opportunity of reviewing this interesting study aime at evaluating the impact on sexual function in young women with thyroid autoimmune disease.
This study has a very interesting topic and has some very important results.
The authors found that the presence of thyroid disease in the female population associates sexual dysfunctionis significantly higher in the group of patients with autoimmune thyroiditis, regardless of the stage of the condition.
The manuscript is overall well written and the topic treated is of major interest.
I have some minor suggestions and questions that I wish the authors could comment on.
-
The authors demonstrated how thyroid autoimmune disease is associated with improved sexual function through the administration of the FSFI questionnaire. It would be very interesting to know how this condition can change other aspects of sexuality. I suggest to evaluate other aspects through the administration of additional questionnaires such as the FSD-R (The Female Sexual Distress Scale-Revised).
-
Patients underwent thyroid evaluation by means of: clinical examination, neck ultra-171 sound and laboratory analysis. Ovarian evaluation was performed by anamnesis – quality of 173 menstruation, quantity, periodicity, clinical examination, normal sexualisation, lack of hy-174 perandrogenic signs and laboratory assays. It would be very interesting to perform an gynecological ultrasound evaluation, specifically an ecocolordoppler study of clitoral artery. A recent study showed that clitoral vascular resistance is positively associated with MetS, decreased sexual arousal, body image concerns, and increased somatized anxiety symptoms. It would be interesting to assess how an autoimmune condition such as thyroiditis can change peripheral vascularity. In this regard the authors may consider to cite this paper in the discussion(PMID: 27692844)
-
This results are extremely interesting. In this regard I suggest to widen the discussion, specifically to deepen the motivation that could be the basis of what this paper show. This aspect needs to be given greater prominence
Author Response
Dear Reviewer,
Thank you for all your relevant comments and generous suggestions.
We attempted to make all corrections addressing the described concerns. Additional language and style editing was performed. Hopefully it will meet your requirements.
Point 1: The authors demonstrated how thyroid autoimmune disease is associated with improved sexual function through the administration of the FSFI questionnaire. It would be very interesting to know how this condition can change other aspects of sexuality. I suggest to evaluate other aspects through the administration of additional questionnaires such as the FSD-R (The Female Sexual Distress Scale-Revised).
Response 1: Thank you for this remark. The FSD-R questionnaire is indeed a valuable tool in diagnosing FSD. We really appreciate your suggestion. However, this is a prospective study, so filling in the FSD-R questionnaire now, retrospectively, is impossible, considering the same study group. We will take into consideration this questionnaire for our future research.
Point 2: Patients underwent thyroid evaluation by means of: clinical examination, neck ultra-171 sound and laboratory analysis. Ovarian evaluation was performed by anamnesis – quality of 173 menstruation, quantity, periodicity, clinical examination, normal sexualisation, lack of hy-174 perandrogenic signs and laboratory assays. It would be very interesting to perform an gynecological ultrasound evaluation, specifically an ecocolordoppler study of clitoral artery. A recent study showed that clitoral vascular resistance is positively associated with MetS, decreased sexual arousal, body image concerns, and increased somatized anxiety symptoms. It would be interesting to assess how an autoimmune condition such as thyroiditis can change peripheral vascularity. In this regard the authors may consider to cite this paper in the discussion(PMID: 27692844).
Response 2: Thank you for sharing this paper and its interesting results. We found a lot of valuable information. We intend to continue our studies in this field and we are really considering your suggestion. Hopefully we will perform this kind of investigation in a further study. We have cited the paper in the discussion (reference no. 35).
Point 3: This results are extremely interesting. In this regard I suggest to widen the discussion, specifically to deepen the motivation that could be the basis of what this paper show. This aspect needs to be given greater prominence.
Response 3: Thank you for your remarks. We have widen the discussion as requested. However, we consider further studies are needed in order to highlight the pathophysiological basis of our findings.
Reviewer 2 Report
Thank you for the possibility to review the manuscript and the interesting results.
The English of the manuscript should be significantly improved.
The statistical methodology for the analyses of the data is problematic and not related to the conclusion drawn. Whenever data is normally distributed the mean and standard deviation should be given otherwise median and range. In the different tables, different values are given. Furthermore, there if independent risk factors needed to be defined, regression analysis should be performed.
I’m also surprised about the significant results that have been found. In relatively small number of participants per group (E, and S) and limited differences in range and mean, highly significant results have been found. I would like to suggest repeating these with the right tests according to the distributions.
Abstract include abbreviation that included the first time and should be written in full words. “Certification” is probably Diagnosis. Please rearrange according to background, methods, results and conclusion.
The conclusion that pains seems uninfluenced by thyroid disease while arousal has been influenced is strange. As lower lubrication often causes dyspareunia. Please explain.
What is the impact of depression on FSD? Maybe this was more of impact on the FSFI than the thyroid abnormality?
What is meant by “Our findings are significant and promising – for what and which purpose.
Introduction:
AITD claimed as common disorder, please provide estimated prevalence and references.
If this study aims “In these conditions, the present paper aims to assess the prevalence of female sexual dysfunction (FSD) in this particular group of women”, then other methodology needed to be chosen. Probably this study aims to estimate the possible comorbidities related to female sexual dysfunction and depression in relation to AITD.
The lines from 103 would be better written that Sexual dysfunctions are highly prevalent among depressive patients and antidepressant use.
Lines 110 to 116 are not relevant for this study
Lines 117-121 and 133-144 are related to methods and not to introduction and could be shorter.
The inclusion criteria should be clarified, specially “compliant with the idea of sexual evaluation”
In the methods: except the comments above: please specify the requirement conditions, the detention of each group (E, C, S) and the number of institutions participated and in which period.
Results:
It stated that general parameters of each group were similar as the control group. Please provide data and the characteristics of the patients.
Line 256 is related to methods and as mentioned above if not normal distribution was obtained all data should be in median and range. (Table 6 and 7).
Again, the significance of the results should be controlled.
Tables 8 to 10 show none significant differences and can be shortly described in text only.
A multiple regression analysis will be probably appropriate to show which component were clearly related to sexual dysfunctions and at which level.
Discussion:
The clinical implications of the results should be discussed with given the relevant references. Furthermore, the relationship between thyroid function, depression and female sexual function could be more elaborated.
With regression analyses the possible independent risk factors may be identified.
Round 2
Reviewer 2 Report
Thank you for the corrections according to previous comments.
Majority of the corrections has been done or appropriate explanation provided.
Two issues are still pending and need to be corrected:
-The sentence in the abstract and conclusion: "Our findings are significant and promising; they may help professionals dealing with sexual and reproductive health" should be corrected according to the answer provided that despite the importance of female sexual dysfunction and its prevalence, clinicians and patients often ignore it. In fact, only a small percentage of patients consult their doctors about sexual health, and their doctors do not often ask them questions related to this aspect.
- The explanation about lubrication issue that may cause dyspareunia was only given in the answer to the reviewer but not included in the discussion ("Response 5: This is a very good point, thank you for pointing it out. The differences are not major. Even if there are some alterations in arousal, they are not so pronounced so that they could influence the pain domain. Lubrication is not entirely absent" ). Please add this issue to the discussion!
